# Utilization of Grain Physical and Biochemical Traits to Predict Malting Quality of Barley (*Hordeum vulgare* L.) under Sub-Tropical Climate

**DOI:** 10.3390/foods11213403

**Published:** 2022-10-28

**Authors:** Dinesh Kumar, Amit Kumar Sharma, Sneh Narwal, Sonia Sheoran, Ramesh Pal Singh Verma, Gyanendra Pratap Singh

**Affiliations:** 1ICAR-Indian Institute of Wheat and Barley Research, Karnal 132001, India; 2ICAR-Indian Agricultural Research Institute, New Delhi 110012, India

**Keywords:** barley, grain quality, malt quality, growing year, correlation

## Abstract

Barley is the most popular raw material for malting, and recently, the demand for malt-based products has increased several folds in India and other South Asian countries. The barley growing season is peculiar in the sub-tropical plains region compared to European or Northern American conditions, characterized by a total crop duration of 130–145 days with a maximum grain filling duration of around only 35–40 days. A total of 19 barley genotypes were grown for three years to assess the comparative performance in relation to different quality traits, including grain physical traits and biochemical and malt quality parameters. Analysis of variance, Pearson correlation, and principal component analysis were performed to determine the correlation among different traits. The results showed significant genotypic variation among genotypes for individual grain and malt traits. Despite the shorter window for grain filling, several good malting genotypes have been developed for the sub-tropical climates. The genotypes DWRUB52, DWRB101, RD2849, DWRUB64, and DWRB91 were found suitable for malting. Based on correlation studies, a few grain parameters have been identified which can be used to predict the malting potential of a barley genotype. The hot water extract was found to be positively correlated with the grain test weight, thousand-grain weight, and malt friability but was negatively correlated with the husk content. Beta-glucan content varied from 3.4 to 6.1% (dwb); reducing the grain beta-glucan content and increasing the amylase could be priorities to address in future malt barley improvement programs under sub-tropical climatic conditions.

## 1. Introduction

Barley (*Hordeum vulgare* L.) is one of the most ancient grains domesticated by humans, probably because of its ability to grow under diverse environments and superior health-benefitting properties. Barley can be grown in diverse climates, ranging from problematic soils to extremes of temperatures, with relatively fewer resource requirements than other cereals. The area planted with barley has decreased over time mainly due to the availability of improved dwarf varieties of wheat that have higher productivity, the development of irrigation infrastructure with assured availability of water, and changing food habits [1]. However, in the past two decades, the area has more or less stabilized. The stabilization in area can be attributed to the increasing industrial use of barley, especially for malt making, as well as the clinical documentation of its health benefits.

Malt is the major industrial product of barley. It is made from specifically bred barley which has several physical and biochemical traits leading to a higher yield and better quality end product [2]. These quality traits are governed by genotype, growing environment, and cultural practices [3]. India and China are predicted to register a very high growth rate in the malting and brewing sector in the future [1]. The availability of the desired quality raw material is an important requirement for setting up any agro-industry to cut down transportation costs and to comply with phytosanitary and other import requirements of importing countries. Many malt barley varieties have been developed that are suitable for spring and winter barley in Europe, Northern America, and Australia, and much information has been generated on the biochemical, physiological, and molecular levels to identify factors affecting malt quality. However, information on these aspects for northern region sub-tropical climates (Figure 1), where the grain filling period is restricted to 35–40 days and experiences relatively higher temperatures [4] with occasional rain, is very scanty. The temperatures start rising after the anthesis and restrict the starch accumulation window in the sub-tropical plains in the Indian subcontinent. The higher temperatures are also expected to affect the polysaccharide biosynthesis and source-sink relationships, ultimately resulting in inferior quality compared to crops being grown under longer duration in temperate climates ([5] and references therein). This reduction in malting quality is mainly attributed to the increase in protein concentrations and decreased accumulation of carbohydrates [5]. Therefore, exhaustive information on malt quality in the barley being produced in sub-tropical climates needs to be generated to supplement the malt barley improvement programs of sub-tropical regions. In a study carried out by Kant et al. [6] using barley grown on hills (1638 m above mean sea level), the effect of a longer growing period with an extended grain filling duration was clearly seen on barley malt quality. Therefore, this study was planned with two objectives: (i) identification of grain and malt traits needing attention from a quality point of view and (ii) to study the correlation among different traits for providing inputs to malt breeding programs.

## 2. Materials and Methods

### 2.1. Grain Samples

A total of 19 genotypes (Table 1) were grown from mid-November to mid-April in Karnal, India (29.7° N and 76.99° E) during 2016–2017, 2017–2018, and 2018–2019 in three replications. The genotypes were selected based on their end usage at one or other time by the Indian malt industry. The crop was fertilized with 90 kg nitrogen (in split), 20 kg phosphorus, and 20 kg potassium, and all other recommended crop management practices, including for weeds, insects/pests, were followed as and when required. The crop was harvested around 10 April every year and thrashed mechanically; the collected grains were cleaned manually and stored in air-tight containers at −20 °C till further analysis.

### 2.2. Flour Preparation

For estimating biochemical parameters, the grains or malt were ground in a Tecator Cyclotec sample mill (Model 1093, FOSS, Hillerød, Denmark) to pass through a 0.5 mm screen. All the analyses on grains, malt, and wort were performed as per the methods suggested by the European Brewery Convention (EBC) [7], and acceptable values were computed using the analytical guidelines for barley breeders in India.

### 2.3. Malt Preparation

Bold/plump grains (grains > 2.5 mm screen) processed on a Sortimat (Pfeuffer make laboratory grader) were used for malting in an automatic micro-malting system (Joe White make, Southbank, VI, Australia). The malting cycle involved steeping, germination, and kilning stages as per the following schedule:Steeping: 8 h dip in water (temperature 18 °C) with continuous aeration → 6 h air rest (temperature 18 °C) → 10 h dip in water (temperature 18 °C) with continuous aeration;Germination: 24 h at 18 °C → 24 h at 17 °C → 24 h at 16 °C;Kilning: 3 h at 45 °C → 3 h at 50 °C → 3 h at 55 °C → 3 h at 60 °C → 3 h at 65 °C → 3 h at 70 °C → 3 h at 75 °C → 3 h at 80 °C.

The malt was removed from the machine after cooling to room temperature, and rootlets were removed by hand rubbing. The malt was then stored in air-tight interlocking polythene bags at −20 °C till further analysis [8].

### 2.4. Grain Physical Traits (Test Weight, Thousand Grain Weight, Bold Grains Percentage)

Test weight was estimated using hectolitre measurement equipment designed by ICAR-IIWBR, Karnal, for small grain samples and weighing up to 1 g accuracy on an electronic balance. The test weight was then expressed as kilogram per hectolitre. For thousand grain weight (TGW), 1000 grains were counted using a Contador (Pfeuffer, Kitzingen, Germany) seed counter and weighed on an electronic balance up to two digits in grams. The grain plumpness was measured using 100-g grains on a Sortimat laboratory grader (Pfeuffer GmbH, Germany) and sieved for 3 minutes using sieves of 2.8 mm, 2.5 mm, and 2.2 mm. The grains retained on sieves 2.5 mm and above were pooled together and called bold/plump grains. The grains that passed through the 2.2 mm sieve were designated as thin grains, while the fraction retained on the 2.2 mm sieve were considered intermediate-size grains [1].

### 2.5. Husk Content

Husk content was estimated on a 20 g grain sample using the sodium hypochlorite method [7].

### 2.6. Protein Content

The protein content was estimated using a near-infrared transmittance (NIR) grain analyzer (Infratech 1241, FOSS, Hillerød, Denmark). The values were expressed on a percent dry weight basis (% dwb).

### 2.7. Beta-Glucan Content

Mixed linkage (1 → 3; 1 → 4)-β-D-glucans were measured using the Megazyme Assay Kit (K-BGLU, Megazyme Ltd., Bray, Ireland) following the method of [9]. Results were expressed as percent on a dry weight basis (% dwb).

### 2.8. Beta-Amylase Activity

The beta-amylase (BA) activity was estimated in grain and malt flour using the Betamyl Assay Kit (Megazyme Ireland Ltd.) as per the procedure of [10]. The activity was expressed as Beta amyl Unit/g.

### 2.9. Malt Friability and Homogeneity

A 50-g quantity of malt was deposited in the friability meter (Pfeuffer, Germany), and the machine was run for 8 min. The malt powder obtained was weighed on an electronic balance to estimate percent friability. This powder, along with the fraction retained in the friability meter, was then mixed and put on the Sortimat (Pfeuffer, Germany) for one minute, and the fraction passing through a 2.2 mm screen was considered homogenous malt.

### 2.10. Wort Preparation

The malt flour was prepared in a Buhler’s laboratory mill (Laboratory Disc Mill, DLFU) at a fine grinding setting, and the flour was extracted in an IEC make (Australia) mashing bath at 45 °C and then at 70 °C for a total duration of 120 min as per EBC procedure [7]. The resulting slurry was used to determine the filtration rate and hot water extract.

### 2.11. Filtration Rate (FR)

The slurry obtained after mashing was filtered through Whatman 2555 ½ filter paper, and the filtrate obtained in one hour was considered the filtration rate (ml/per hour).

### 2.12. Hot Water Extract (HWE)

The hot water extract or malt extract was determined using Borosil make A grade-specific gravity bottles. Fifty ml of wort was kept at 18 °C for 20 min, and specific gravity was measured. The hot water extract or malt extract was computed from the standard EBC table and expressed as percent fgdwb (fine ground dry weight basis).

### 2.13. Statistical Analysis

Data are reported as mean ± standard deviation for three determinations of each sample. For all the data, the effects of the genotype (G), year (Y), and G × Y interactions were calculated via a combined analysis of variance (ANOVA) using the Statistical Analysis System (SAS version 9.3 for Windows, SAS Inc., Chicago, IL, USA) software package. Statistical comparisons were significant when *p* < 0.05 (*), *p* < 0.001 (**), or *p* < 0.0001 (***). Tukey’s comparison test (*p* ≤ 0.05) was also performed using SAS software 9.3 to identify differences between the values. A correlation analysis was carried out in Rstudio using the ‘corrplot’ package (RStudio^®^, Boston, MA, USA). Principal component analysis (PCA) was conducted using JMP (trial version, SAS Corp., Cary, NC, USA).

## 3. Results

The grain and malt quality parameters are governed by both the genotype and the environment. The quantity and quality of the malt are determined by the quality of barley grains. A set of nineteen barley genotypes were grown for three years under a sub-tropical climate and analyzed for eight grain quality and five malting quality parameters. The three-year cumulative data for different grain and malt quality traits is provided in Table 2. The effect of growing year on the grain and malt quality traits is shown in Table 3 and Table 4, respectively. The effect of genotype, environment, and their interaction on different quality parameters is shown in Table 5. The association between different quality traits was analyzed by Pearson correlation (Figure 2) and PCA (Figure 3).

### 3.1. Grain Quality Traits

The grain protein content is a very important trait, and generally, a range of 9.0–11.0% (dwb) is considered optimum for malting purposes. The protein content ranged from 9.8% (DWRUB 64 and RD 2552) to 12.4% (RS 6). Ten genotypes had protein content in the optimum range for malting. There was a significant effect of the growing season on grain protein content (*p* < 0.0001). The highest protein content was obtained in 2018–2019, and the lowest was in 2017–2018.

The physical grain parameters: test weight, TGW, and bold grain percentage, were significantly affected by both genotype and growing year (Table 5). Test weight or hectolitre weight is a measure of grain density, and higher test weight is related to higher-end product recovery; values of 65 kg/hl or more are considered optimum. The test weight values varied from 59.9 kg/hl (DL 88) to 69.4 kg/hl (DWRUB 52). Out of 19 genotypes tested, 7 had a test weight of 65.0 kg/hl or more. The thousand-grain weight varied from 38.0 g (NDB 1173) to 57.0 g (DWB 91). The weight of individual kernels has important implications on grain yield and quality. A TGW value of 42 to 46 g is considered optimal for malting purposes. Too low values indirectly indicate lower starch content and higher husk content; on the other hand, very high thousand-grain weight leads to excessive large-sized grains, which may cause problems in proper malting. If the grain size is bigger, the imbibition of water to the core of the endosperm is not sufficient, and therefore, the degradation of biomolecules remains incomplete, leading to lower malt extract values. Six genotypes were found in the ideal range. The bold grain percentage is one of the most important grain parameters as it indirectly represents the starch content in the endosperm. More plump grains have the best quality of the malt and lead to higher hot water extract values. For malt barley, genotypes with >90% plump grains are preferred. Normally, two-rowed genotypes have a higher bold grain percentage compared to six-rowed barleys, and therefore, two-rowed varieties are preferred for malt making [11]. The bold grain percentage varied from 61.1% (RD 2668) to 95.0% (DWRB 92), with statistically significant differences among genotypes and growing seasons. The genotypes with >90% plump grains were DWR 28, DWRB 91, DWRUB 64, and DWRB 92. However, the genotype DWRUB 64, despite being a six-rowed cultivar, had a higher bold grain percentage (91.4%). The percentage of thin grains should not be more than 3% in any malt variety. Nine genotypes were found to have less than 3% thin grain percentage in this study.

Beta-glucans contribute approximately 75% to the endosperm cell walls, and therefore, higher grain beta-glucan content is considered highly undesirable since thicker walls lead to poor and delayed endosperm modification during malting. The beta-glucan content ranged from 3.3% (K 551) to 6.1% (DWR73), with statistically significant differences among the genotypes and growing years. A grain beta-glucan content of 4.0% or less is considered suitable for any malt variety, and varieties K 551 (3.3%), Amber (3.4%), and BCU 73 (4.0%) have desirable values.

Adhered husk protects the growing acrospire during the malting process, and the spent grains also act as a filter bed during the lautering process. Though husk content is important in malting barley, its higher content is not desirable as it leads to a decrease in end-product recovery. A husk content of <10.5% is considered the most desirable for malting. The husk content varied from 10.1% (Alpha 93) to 13.0% (Amber). Only two genotypes, Alpha 93 and DWRB 101, recorded 10.1% and 10.4% husk content, respectively. The husk content was influenced by the genotype, but no significant effect of the growing year was observed on the husk content.

Four enzymes contribute to diastatic power, but alpha and beta amylases are considered the major ones. Alpha amylases are undetectable in the raw grain, and their activity is noticed after the malting process/germination is started. Beta-amylase is usually considered the most important enzyme contributing to diastatic power because it typically has the highest activity of all the barley endosperm amylolytic enzymes [12]. Among the nineteen varieties studied, the grain beta-amylase activity varied from 13.8 beta amyl units/g flour in DWRB 52 to 25.8 beta amyl units/g flour in DWRB 92. Six genotypes showed beta-amylase activity of 20 amyl units/g or more. The genotype and the growing year both had significant effects on grain beta-amylase activity, but the interaction between the genotype and the growing year had no influence on the beta-amylase activity.

### 3.2. Malt Quality Traits

The quality of the malt mainly depends on the quality of the barley grains and the malting conditions. The quality of the malt ultimately defines the quality of the different end products. Friability is a measure of the degree of endosperm modification; higher values indicate a better breakdown of biomolecules during the malting process. The percent friability varied from 48% (RS 6) to 71.6% (DWRUB 64), with a statistically significant effect of both genotype and growing season. A friability percentage of 70 or more was obtained in Alpha 93, DWRB 101, DWRUB 52, DWRB 91, and DWRUB 64. The malt powder obtained from the friability meter should be homogenous, with a minimum content of big-size fractions. The malt homogeneity varied from 75.0% in Clipper to 95.9% in DWRB 91. Malt homogeneity of 90% or more was obtained in seven genotypes. The malt homogeneity was influenced by the genotype but not by the growing year and genotype × year interaction.

Beta-amylase activity in malt plays an important role in starch breakdown during mashing, and genotypes with higher beta-amylase activity are considered more suitable for malting. The malt beta-amylase activity ranged from 8.4 beta amyl units/g malt flour (DWRUB 52) to 25.4 beta amyl units/g malt flour (RS 6). Three genotypes, RS 6, DWRB 92, and DWR 28, were found to have malt beta-amylase activity of more than 20 beta amyl units/g malt flour. Malt beta-amylase activity was significantly influenced by both the genotype and growing year but not by genotype × year interaction. The malt beta-amylase was significantly lower during 2017–18 (Table 4).

The filtration rate is one of the important criteria for the selection of barley for malting. A higher filtration rate is considered desirable as it saves time and the end product’s quantity and quality are better. The values ranged from 226.1 mL/h (DWRB 91) to 283.9 mL/h (Amber), with nine genotypes giving desirable values of 250 mL/h or more. The filtration rate was significantly affected by the genotype but not by the growing year. Hot water extract (HWE) is the penultimate indicator of the malting potential of a genotype with respect to end-product recovery. However, in addition to the percentage of extract, its composition also plays an important role in the requirements of different industries. Though the hot water extract values numerically varied from 76.3% (Amber) to 81.0% (DWRB 91), the differences were statistically non-significant. A hot water extract of 80% or more is considered desirable for any malt variety (especially in two-rowed ones). This benchmark was obtained in DWRUB 64, DWRUB 52, DWRB 101, and DWRB 91. DWRUB 64 is a six-rowed variety and has a hot water extract value of 80.1%, which is exceptional. Higher HWE value in DWRB 91, along with higher grain beta-glucan content, points to the role of several other factors governing the final malt extract values.

### 3.3. Effect of Growing Year

A number of malting quality traits are influenced by environmental conditions such as temperature, rainfall, and soil type. The meteorological data for the grain filling period (10 February to 27 March coinciding with the general grain filling duration) is provided in Table 6. As per the meteorological data during the grain filling period, there was a sudden increase in the temperature late in the grain filling period during 2016–2017. During 2018–2019, the temperatures were high throughout the entire grain-filling period. The hot and dry period during 2018–2019 corresponded to the period of cell division in starchy endosperm. Such conditions shortened the length of this period, thus influencing the accumulation of starch. For this reason, the season of 2018–2019 resulted in high protein content (13.3%) and lower starch content, as indicated by the reduced bold grain percentage (77.7%) (Table 3). Due to higher protein content, the beta-amylase was also high in these two years. Beta-glucan content was also the highest during 2018–2019 (5.2%). Combining all these values with the high temperatures, negligible rain, and less sunshine, the lowest hot water extract percentage was obtained in 2018–2019 (75.7%) (Table 4 and Table 6). Conversely, during 2017–2018, there was good precipitation during the early grain filling period leading to comparatively lower temperatures during the entire grain filling duration. These conditions were found congenial for good grain filling and optimum malt production. During 2017–2018, the beta-amylase activity was less, which was mainly due to the low protein content (9.0%). However, all other grain and malt quality parameters were found to be optimum during this year. These results clearly indicate that the environmental conditions during a growing season play a significant role in determining the malting quality of barley with different genotypes.

### 3.4. Correlation among Different Grain and Malt Quality Traits

A total of 13 traits were analyzed in this study, and certain positive and negative correlations were observed between different grain and malt quality traits (Figure 2). The hot water extract (HWE) is the most important indicator of the quality of malt produced from barley grain from the malting and brewing industry point of view [13]. HWE showed positive correlation with test weight (0.62), TGW (0.47), bold grain percentage (0.40), malt friability (0.70), and homogeneity (0.74). The malt friability was found to be positively correlated to bold grain percentage (0.54) and negatively correlated to husk content (−0.65). Malt homogeneity showed a high positive correlation (0.73) with malt friability. The filtration rate showed a high negative correlation with beta-glucan content (−0.74). Malt BA had a high positive correlation with grain protein content (0.60) and grain BA (0.87).

### 3.5. Principal Component Analysis (PCA)

Malting quality is complex and multidimensional; therefore, principal component analysis can be a useful method to study the association between traits [14]. PCA separated the genotypes based on 13 studied traits (Figure 3). Out of the 13 principal components (PCs) extracted from the original variables, the first 2 PCs explained 60.8% of the total variation. The amount of variation explained by PC1 and PC2 were 42.6% and 18.2%, respectively. TGW and grain beta-amylase showed the highest vector positive loadings, whereas friability and thin grains (%) showed the highest negative vector loading with PC1 and PC2. These results from PCA further demonstrate that all the genotypes used in the present study had substantial genetic diversity for all 13 traits. Genotypes DWR28, Alpha93, DWRB73, DWRB91, and Clipper were aligned with beta-glucan content, TGW, test weight, HWE, and bold grain percent.

## 4. Discussion

The malting potential of barley depends on the biochemical attributes of the mature grain, which, in turn, are influenced by both the genotype and environmental factors such as temperature, rainfall, nitrogen fertilizer, and soil type [15]. This is especially important in sub-tropical regions, where the high temperatures during and after the grain-filling period may limit the starch accumulation window. This leads to grain quality that is considered inferior to the grain quality of temperate regions. Therefore, growing barley with good malting quality is a challenge under sub-tropical conditions.

In order to establish a correlation between the grain and malt quality traits under Indian conditions, results from the analysis of nineteen genotypes for eight grain quality and five malt quality traits for three growing seasons were used. PCA and correlation analysis revealed that most of the quality traits are highly interrelated, either positively or negatively. Many studies have reported correlation coefficients between individual traits of grain and malting quality, but the results are quite different depending on geographical region, selection of genotypes, seasons, and crop management [14,16]. The combined analysis of variance indicated a significant effect of genotype and growing year on most of the studied traits, except husk content and filtration rate, for the growing year. The interaction between the genotype and the growing year was found significant, albeit to different levels, except for homogeneity, grain, and malt beta-amylase activity, indicating the effect of growing year conditions on different genotypes. Molina–Cano et al. [17] revealed contradictory results about the existence and magnitude of the genotype x environment interaction. Generally, the environment has more influence on total variation than the interaction with genotypes.

Each growing year is characterized by a specific set of environmental conditions, such as temperature, humidity, rainfall, and sunshine. It is reported that mild heat stress late in the grain-filling period results in decreased grading [18]. High temperatures and moisture stress can limit the amount of grain fill due to reduced starch metabolism. The accumulation of starch is reported to be more sensitive to high temperatures than to the accumulation of nitrogen, resulting in increased grain nitrogen proportion and higher protein contents [19]. In the present study also, the high temperatures during the grain filling period of 2018–2019 resulted in a lower bold grain percentage, indicating less starch deposition, the highest protein content, high beta-glucan content, and the lowest HWE percentage. In sub-tropical environments, such conditions are a regular feature resulting in the low malting quality of the barley genotypes as compared to cool temperate regions.

In the last few years, there has been continuous demand from industry for higher protein malts, primarily for use in malt-based food products. There is a significant correlation between grain protein content and malt diastatic power [20]. Beta-amylase is usually considered the most important enzyme that contributes to diastatic power because it typically has the highest activity of all of the barley endosperm amylolytic enzymes [1]. The present study also showed a high positive correlation between the protein content and the grain and malt beta-amylase activity. Such kinds of barley may be used for brewing unmalted cereals (adjuncts), as barley malt can act as a source of diastatic enzymes. Thus, there is a need for barley genotypes with a protein content of 12–14% without much compromise on other grain and malt quality parameters [1]. There was a significant effect of the growing season on grain protein content. It is reported in the literature that, in addition to the genotype, growing conditions and cultural practices also affect the grain protein content [21]. Thus, the stability of this trait needs to be strengthened under Indian growing conditions to meet the demands of the malt industry. Beta-amylase is of great importance in producing the substrates (i.e., the fermentable sugars glucose, maltose, and maltotriose) for fermentation by yeast. Beta-amylase activity correlates with fermentable sugar production during mashing to a much greater extent than any other diastatic power enzyme in malt [22]. Further in-depth studies are required on this enzyme, especially in relation to grain quality traits under Indian growing conditions.

Higher grain beta-glucan content is considered highly undesirable for malting since thicker walls lead to poor and delayed endosperm modification during malting. Ram [23] reported a large variation in grain beta-glucan content in the barley genotypes ranging from 2.8 to 7.1%. They also reported a wort filtration range from 80 mL/h to 335 mL/h under Indian conditions and found a significant negative correlation between filtration rate and grain beta-glucan content. Most of the genotypes showed high beta-glucan content (>4.0%) in the present study. The high beta-glucan content increases the viscosity of the wort and thus hampers the filtration rate. The higher filtration rate is considered desirable as it saves time and the end product’s quantity and quality are better. Therefore, there is a need to breed for barley genotypes with low beta-glucan, especially for malting purposes.

Hot water extract or malt extract is the penultimate indicator of the potential of a malt variety with respect to end-product recovery. However, in addition to the percentage of extract, its composition also plays an important role as per the requirements of different industries. Composition, in turn, depends on the activity of different amylolytic enzymes during the mashing process. The complex nature of HWE is the product of many physical and biochemical factors affected by genotype, environment, cultural practices, and malting conditions. DWRUB 64 is a six-rowed variety and has a hot water extract value of 80.1%, which is exceptional. Higher HWE value in DWRB 91 despite exceptionally higher thousand-grain weight and higher grain beta-glucan content points to the role of several other factors governing the final malt extract values [1]. The HWE was found to be positively correlated with the test weight, TGW, friability, and homogeneity but was negatively correlated with the husk content. This is because a high amount of starch is available in grains with high test weight and TGW, and also because of the good friability and homogeneity, the enzyme accessibility increases in converting the starch to the fermentable sugars. Verma [24] studied the malting quality of 72 Indian barley varieties and found a positive correlation between test weight and hot water extract. The inverse relationship between malt extract and grain protein content is determined by the variety and environment, and therefore, the value of the relationship is more or less specific for each variety. This negative correlation is mainly due to the hordein fraction of the proteins. The hordein fraction has two main negative effects: decreasing starch levels due to a negative correlation with starch and restricting access by amylolytic enzymes to starch during germination since the starch granules are embedded into the endosperm protein matrix with hordein as the main component [25].

The study provided very important information that the varieties DWRB 91 (2-rowed) and DWRUB 64 (6-rowed), both released for late sown conditions, have excellent quality when sown in timely conditions. The varieties DWRUB 52, DWRB 101, and RD 2849 are good malt barley varieties except for the drawback of relatively lower diastatic power (beta-amylase activity). DWRB 92 could prove to be an excellent raw material for malt making, provided the steeping conditions are modified. Genotypes DWR 28, Alpha 93, DWRB 73, DWRB 91, and Clipper were aligned together with beta-glucan, TGW, test weight, HWE, and bold grain (%) by PCA. An important point from this study could be achieving higher diastatic power with higher extract value; however, higher beta-amylase and lower grain beta-glucan traits need to be addressed. Some progress in this direction has been reported [26,27]. Despite a short grain filling window of 30–40 days, which is subject to fluctuating temperatures, marvelous achievements have been made by the Indian malt barley program by breeding for quality without compromising the grain yield and biotic/abiotic stress tolerance [28].

Measuring HWE is a tedious, expensive, and time-consuming process [29]. Correlation analysis has shown that one can reduce the complexity of malting quality parameters, and some highly correlated parameters may be chosen for easy measurements, especially the grain traits, for the prediction of HWE. In a study by Verma et al. [30] on Indian and exotic genotypes grown at two locations in the northwestern plains of India, hectolitre weight, thousand-grain weight, hull content, and malt friability were used as selection criteria for superior malting quality in early generations of a malt barley improvement program. A similar trend is obtained in this study, but a positive correlation between these grain traits with beta-glucan content has given the thread for a further in-depth investigation on this aspect. Though this study was conducted with a very small number of genotypes, grain beta-glucan and grain beta-amylase activity could be another supplementing factor for an early selection, where the quantity of grains is a limitation.

## 5. Conclusions

Barley quality traits are quantitatively inherited and highly influenced by environmental factors such as temperature, available water, nitrogen fertilizer, and soil type. This three-year study was carried out to study the effects of genotype and growth year on different grain and malt quality traits in barley grown under sub-tropical conditions where the grain filling periods are relatively short. The temperatures start rising after the anthesis and restrict the starch accumulation window. This leads to the inferior quality of grains in comparison to temperate regions. The growing season significantly affected most of the grain and malt quality traits except the husk content of the grain, malt homogeneity, and wort filtration rates. The interaction between the genotype and the growth year was found significant for all the studied traits except homogeneity, grain beta-amylase, and malt beta-amylase, showing the effect of climatic conditions on different genotypes. Hot water extract is certainly the most important indicator of malt barley quality. The HWE was found to be positively correlated with the test weight, TGW, friability, and homogeneity but was negatively correlated with the husk content. Beta-glucan content is also a very important trait from a malting point of view and is significantly influenced by both the genotype and the growing year. Malt beta-amylase activity is very important for brewing and is highly correlated with grain protein content. Most of the traits studied showed strong positive or negative correlations with each other, and it becomes very difficult to combine all the desirable traits in one single genotype. In fact, this has been the limitation of the malt barley breeding program. Therefore, an optimum level of most of these traits should be combined while making some compromises. Correlation analysis has shown that the complexity of malting quality parameters can be reduced, and some highly correlated parameters may be chosen for easy measurements, especially the grain traits, for the prediction of hot water extract.

## Figures and Tables

**Figure 1 foods-11-03403-f001:**
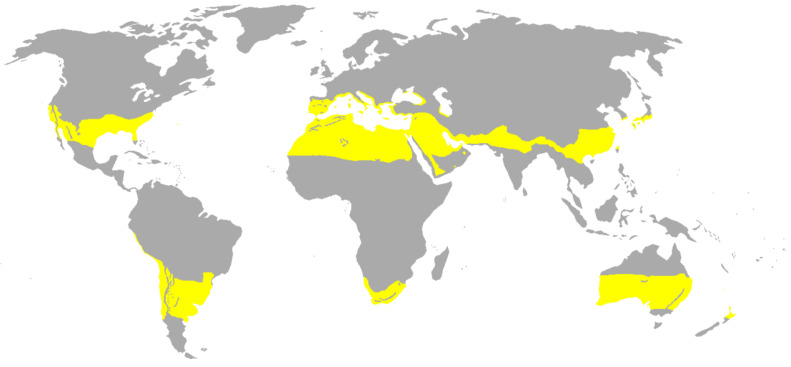
Areas of the world with subtropical climates (in yellow) according to Köppen climate classification (Source/Reproduced from: https://en.wikipedia.org/wiki/Subtropics#/media/File:Subtropical.png under fair use clause; (accessed on 27 September 2022).

**Figure 2 foods-11-03403-f002:**
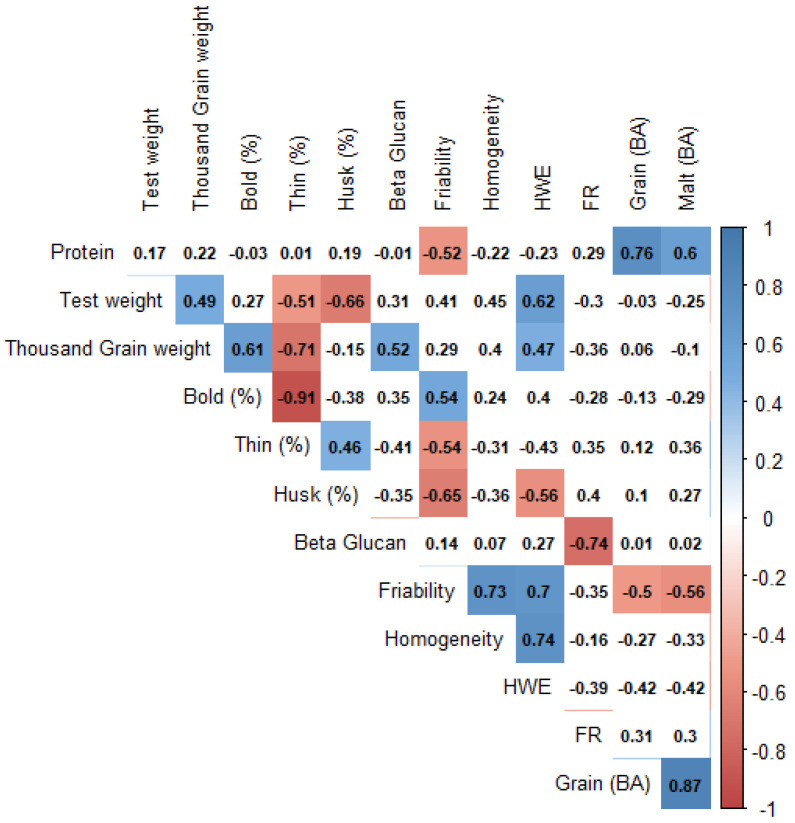
Correlation among different grain and malt quality traits.

**Figure 3 foods-11-03403-f003:**
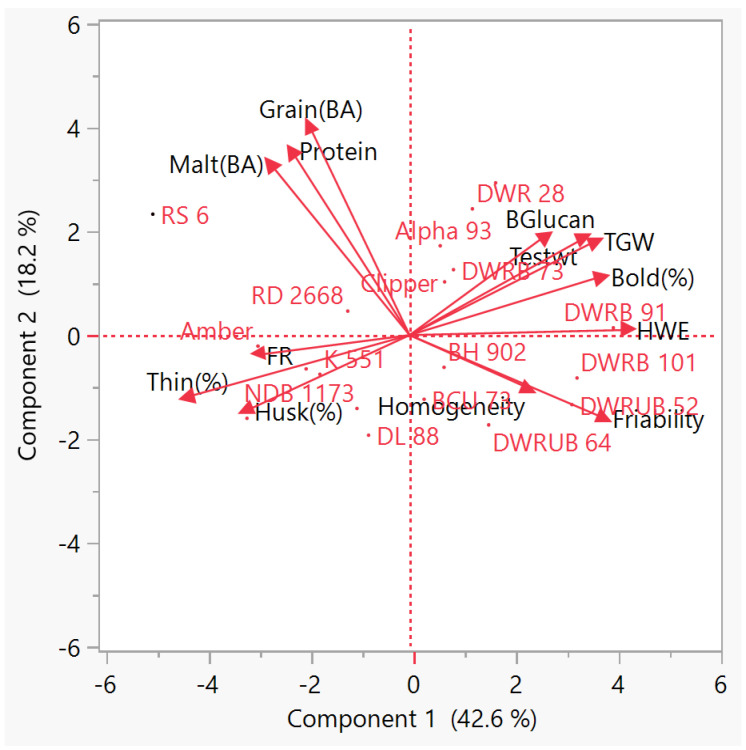
Principal component analysis (PCA) of barley genotypes (in red font) based on the 13 traits (in black font) (variables).

**Table 1 foods-11-03403-t001:** Details of genotypes used in the study.

S. No.	Variety	Year of Release	Row Type	Average Yield (q/ha)	End Use
1	Alfa 93	1995	2	30.00	Malt
2	Amber	1978	6	17.80	Feed
3	BCU 73	1997	2	32.80	Malt
4	BH 902	2010	6	49.75	Feed
5	Clipper	1969	2	30.00	Malt
6	DL 88	1998	6	34.72	Malt
7	DWR 28	2002	2	41.40	Malt
8	DWRUB 52	2007	2	45.10	Malt
9	DWRB 92	2014	2	49.81	Malt
10	DWRB 101	2015	2	50.10	Malt
11	DWRB 73	2011	2	38.70	Malt
12	DWRB 91	2013	2	40.62	Malt
13	DWRUB 64	2012	6	40.50	Malt
14	K 551	1998	6	37.36	Malt
15	NDB 1173	2005	6	35.20	Feed
16	RD 2552	2000	6	44.06	Feed
17	RD 2668	2007	2	42.50	Malt
18	RD 2849	2016	2	50.90	Malt
19	RS 6	1978	6	27.50	Feed

**Table 2 foods-11-03403-t002:** Grain and malt quality traits of different barley genotypes.

	Grain Traits								Malt Traits				
	Protein (% dwb)	Test Weight (kg/hl)	TGW (g)	Bold Grains (%)	Thin Grains (%)	Husk (%)	β-Glucan (%)	Grain BA (Beta Amyl Units/g)	Friability (%)	Homogeneity (%)	Malt BA (Beta Amyl Units/g)	Filtration Rate (mL/h)	HWE (%)
Alpha 93	11.5 ^a^	66.8 ^bdac^	40.9 ^fe^	83.3 ^bda^	3.7 ^bac^	10.1 ^c^	4.3 ^f^	23.7 ^ab^	70.6 ^a^	94.8 ^ab^	19.4 ^bcd^	268.3 ^ab^	78.3 ^a^
Amber	11.1 ^a^	62.1 ^gfhe^	43.7 ^fde^	69.0 ^ed^	5.4 ^bac^	13.0 ^a^	3.4 ^gh^	21.4 ^abcde^	53.5 ^bc^	87.9 ^abcde^	17.5 ^cdef^	283.9 ^a^	76.3 ^a^
BCU 73	10.7 ^a^	65.4 ^bdfc^	49.9 ^bdac^	73.6 ^ebda^	5.4 ^bac^	11.6 ^abc^	4.0 ^fgh^	16.3 ^efghi^	65.0 ^ab^	92.6 ^abcd^	12.6 ^fg^	255.0 ^ab^	79.1 ^a^
BH 902	10.2 ^a^	62.2 ^gfhe^	45.0 ^fde^	90.3 ^ba^	2.3 ^bac^	11.6 ^abc^	5.4 ^abcd^	13.9 ^i^	62.5 ^abc^	80.9 ^f^	12.9 ^fg^	261.1 ^ab^	77.5 ^a^
Clipper	11.5 ^a^	65.1 ^dfce^	41.7 ^fe^	84.5 ^ba^	2.6 ^bac^	10.7 ^abc^	4.5 ^ef^	20.8 ^abcdef^	57.7 ^abc^	75.0 ^f^	14.7 ^def^	245.6 ^ab^	76.4 ^a^
DL 88	10.3 ^a^	59.9 ^h^	40.6 ^fe^	81.2 ^ebda^	4.4 ^bac^	12.1 ^abc^	4.5 ^ef^	14.8 ^ghi^	67.1 ^ab^	89.1 ^abcd^	12.8 ^fg^	235.6 ^b^	77.4 ^a^
DWR 28	11.9 ^a^	64.5 ^gdfce^	53.8 ^ba^	91.0 ^ba^	1.3 ^c^	12.1 ^abc^	5.6 ^abc^	22.7 ^abcd^	62.6 ^abc^	89.3 ^abcd^	21.3 ^abc^	246.1 ^ab^	77.4 ^a^
DWRUB 52	10.8 ^a^	69.4 ^a^	49.2 ^bdc^	83.7 ^ba^	1.8 ^bc^	11.1 ^abc^	4.7 ^def^	13.8 ^i^	71.0 ^a^	93.1 ^abc^	8.4 ^g^	242.2 ^ab^	80.3 ^a^
DWRB 92	11.8 ^a^	65.5 ^bdfce^	54.7 ^ba^	95.0 ^a^	0.5 ^c^	10.9 ^abc^	5.2 ^bcde^	25.8 ^a^	62.4 ^abc^	86.7 ^bcde^	23.4 ^ab^	250.0 ^ab^	78.9 ^a^
DWRB 101	10.5 ^a^	69.0 ^ba^	45.7 ^dec^	84.7 ^ba^	2.0 ^bc^	10.4 ^bc^	4.6 ^def^	14.6 ^ghi^	70.7 ^a^	95.2 ^a^	12.6 ^fg^	250.0 ^ab^	80.6 ^a^
DWRB 73	11.6 ^a^	65.9 ^bdae^	52.4 ^bac^	87.5 ^ba^	2.4 ^bac^	11.5 ^abc^	6.1 ^a^	18.4 ^cdefghi^	58.4 ^abc^	89.9 ^abcd^	13.6 ^efg^	226.7 ^b^	79.0 ^a^
DWRB 91	10.7 ^a^	67.5 ^bac^	57.0 ^a^	91.2 ^ba^	0.8 ^c^	10.6 ^bc^	6.0 ^ab^	17.1 ^efghi^	71.2 ^a^	95.9 ^a^	14.9 ^def^	226.1 ^b^	81.0 ^a^
DWRUB 64	9.8 ^a^	63.3 ^gdfhe^	45.2 ^fdec^	91.4 ^ba^	1.52 ^c^	11.1 ^abc^	4.8 ^cdef^	15.6 ^fghi^	71.6 ^a^	91.1 ^abcd^	13.6 ^efg^	238.9 ^ab^	80.1 ^a^
K 551	11.3 ^a^	63.5 ^gdfhe^	45.0 ^fde^	81.8 ^ebda^	4.5 ^bac^	12.2 ^abc^	3.3 ^h^	17.5 ^defghi^	61.9 ^abc^	86.0 ^cde^	16.7 ^cdef^	262.8 ^ab^	78.00 ^a^
NDB 1173	10.4 ^a^	63.7 ^gdfhe^	38.0 ^f^	75.3 ^ebda^	5.7 ^bac^	10.7 ^abc^	4.5 ^ef^	16.2 ^efghi^	63.7 ^abc^	89.9 ^abcd^	15.0 ^def^	247.8 ^ab^	79.1 ^a^
RD 2552	9.9 ^a^	61.0 ^gh^	38.5 ^fe^	70.8 ^ebd^	8.8 ^ba^	12.1 ^abc^	4.6 ^def^	18.2 ^cdefghi^	61.0 ^abc^	84.7 ^de^	19.4 ^bcd^	247.2 ^ab^	77.2 ^a^
RD 2668	10.6 ^a^	65.8 ^fde^	43.6 ^ef^	61.1 ^e^	6.5 ^bac^	10.9 ^abc^	5.8 ^ab^	19.9 ^bcdefg^	61.3 ^abc^	86.0 ^cde^	18.8 ^bcde^	228.4 ^b^	77.6 ^a^
RD 2849	11.4 ^a^	61.8 ^gfh^	44.2 ^fde^	75.3 ^ebda^	6.8 ^bac^	11.6 ^abc^	4.1 ^fg^	19.2 ^bcdefg^	64.2 ^ab^	90.2 ^abcd^	17.2 ^cdef^	268.9 ^ab^	78.9 ^a^
RS 6	12.4 ^a^	63.5 ^gdfhe^	38.8 ^fe^	61.5 ^e d^	9.1 ^a^	12.6 ^ab^	4.5 ^ef^	23.4 ^abc^	48.0 ^c^	80.2 ^f^	25.4 ^a^	263.9 ^ab^	77.6 ^a^

Different letters indicate significant differences (*p* ≤ 0.05).

**Table 3 foods-11-03403-t003:** Effect of growing year on grain quality traits.

Crop Year	Protein (% dwb)	Test Weight (kg/hl)	TGW (g)	Bold Grains (%)	Thin Grains (%)	Husk (%)	β-Glucan (%)	Grain BA (Beta Amyl Units/g)
2016–2017	10.6	64.5	42.2	72.6	5.7	11.3	4.5	18.4
2017–2018	9.0	65.1	49.5	91.6	1.2	11.6	4.6	15.7
2018–2019	13.3	64.0	45.4	77.7	5.0	11.4	5.2	21.6
LSD (5%)	0.4	0.6	0.9	2.5	1.1	0.4	0.1	0.6

**Table 4 foods-11-03403-t004:** Effect of growing year on malt quality traits.

Crop Year	Friability (%)	Homogeneity (%)	Hot Water Extract (%)	Filtration Rate (mL/h)	Malt BA (Beta Amyl Units/g)
2016–2017	61.6	89.0	79.0	249.7	17.5
2017–2018	69.1	88.0	80.7	246.2	13.8
2018–2019	59.5	88.1	75.7	253.9	17.7
LSD (5%)	2.5	1.5	0.9	9.3	0.9

**Table 5 foods-11-03403-t005:** Combined analysis of variance for nineteen genotypes for the studied traits.

	Mean Squares (MS)		
Traits	Genotypes (G)	Year (Y)	Genotype × Year
Protein	4.69 ***	267.22 ***	1.95 *
Test weight	60.07 ***	18.55 **	12.71 ***
TGW	292.17 ***	762.49 ***	14.14 **
Bold grains	905.81 ***	5519.8 ***	239.26 ***
Thin grains	60.99 ***	331.91 ***	20.09 **
Husk	5.87 ***	1.18	3.39 **
β-Glucan	5.60 ***	8.06 ***	0.26 **
Friability	355.97 ***	1441.03 ***	141.91 ***
Homogeneity	266.17 ***	16.48	17.84
HWE	16.98 **	365.50 ***	16.12 ***
FR	2182.87 ***	846.67	1578.3 ***
Grain BA	115.66 ***	493.78 ***	4.04
Malt BA	159.60 ***	271.30 ***	8.23

Significant at level * *p* < 0.05; ** *p* < 0.001; *** *p* < 0.0001. TGW-Thousand grain weight; HWE-Hot water extract; FR- Filtration rate; BA-Beta-amylase activity.

**Table 6 foods-11-03403-t006:** Meteorological information for the grain filling period of three growing years.

Julian Weeks	Max Temperature (°C)			Min Temperature (°C)		
	2016–2017	2017–2018	2018–2019	2016–2017	2017–2018	2018–2019
12–18 Feb	23.8	21.4	23.7	8.4	8.6	9.9
19–25 Feb	25.0	24.7	27.7	9.9	10.4	12.6
26–04 Mar	26.3	26.8	30.0	9.3	13.0	13.7
05–11 Mar	24.8	27.5	33.9	10.1	11.0	16.8
12–18 Mar	23.4	29.6	35.4	7.6	12.7	18.4
19–25 Mar	30.3	29.5	33.0	13.7	13.7	18.6
	**Rainfall (mm)**			**Sunshine (Hrs/Day)**		
	**2016–2017**	**2017–2018**	**2018–2019**	**2016–2017**	**2017–2018**	**2018–2019**
12–18 Feb	0	29.0	2.0	7.5	7.9	2.6
19–25 Feb	0	0	0	8.1	7.6	4.5
26–04 Mar	0.3	0	0	8.8	8.0	4.1
05–11 Mar	7.5	0	0	8.4	9.7	5.6
12–18 Mar	0	0	0	9.2	10.3	7.2
19–25 Mar	0	0	1	10.0	10.3	6.1

Data taken from Progress Report of AICRP on Wheat and Barley, Barley Network, ICAR-IIWBR, Karnal, Haryana, India for 2016–2017, 2017–2018, 2018–2019.

## Data Availability

The data presented in this study are available in the Annual Reports of ICAR-Indian Institute of Wheat and Barley Research, Karnal–132 001, India. The Figure 1 is available in public domain https://en.wikipedia.org/wiki/Subtropics#/media/File:Subtropical.png (accessed on 27 September 2022).

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
