# Peer review of "Utilization of Grain Physical and Biochemical Traits to Predict Malting Quality of Barley (Hordeum vulgare L.) under Sub-Tropical Climate"

_foods, 2022, doi:10.3390/foods11213403_

Round 1

Reviewer 1 Report

As a researcher in the field of farming, I am very interested in your work. I have looked thoroughly at your article and I see that you did a lot of work on it.

However, There are some problems in the article that need to be solved, if I understand your description correctly. As far as I see, the paper can be considered if the points below are dealt with appropriately.

Specific comments

Abstract

1. There is an extra space before "The" in line 10 and "130-145" in line 12. You are advised to delete it.

2. The abstract includes only the results of the analysis of genotypes and without the results of the analysis of physical and biochemical traits, it is suggested to add.

3. Lack of data support

Introduction

4. The Latin scientific name for barley is missing a dot, please add it.

5. Please add a reference after “The area of barley had decreased over the time mainly due to availability of improved dwarf varieties of wheat with higher productivity, development of irrigation infrastructure with assured availability of water and changing food habits.”

6. There is a redundant space in line 40 for "[2]", please delete.

7. Please add references to lines 45 and 46.

8. The unit linking symbol "/ or -1" is missing from line 59 in "(1638 m msl)", please add.

9. Line 40-41: “India and China have been predicted to register very high growth rate in malting and brewing sector in the future.” lack of literature.

10. Line 51-53: “The higher temperatures are also expected to affect the polysaccharide biosynthesis and source sink relationships ultimately resulting into inferior quality as compared to the crops being grown under longer duration temperate climates.” lack of literature.

Materials and Methods

11. Specific methods and significant differences (0.05/0.01) in the correlation analysis are missing from Table 1, please add.

12. Correlation analysis not performed in Table 2, please add.

13. Please add references to related methods in 2.3, 2.6, 2.7 2.10.

14. In 2.12 Table I is repeated from the previous section, please mark in order.

15. Line78: Between the number and the unit requires a unit of space, such as 78 lines -20 °C, pay attention to the full text of unity.

16. Line 148: recheck and modify the table number, full text check.

Result

17. Data relating to the interaction of genotypes with the environment as mentioned in the text do not appear in Table 4.

18. Figure 1 is not a Pearson correlation analysis, please amend.

19. The year "2018-19" is inappropriate and should read "2018-2019", please amend this in full.

20. In line 190 there is a space before "These conditions", please delete.

21. In 3.2, when describing the analysis of results about genotypes, please label which table or graph it is.

22. In 3.3, when describing the analysis of the results concerning malt quality traits, please label which table or graph it is.

23. The content of lines 288 and 289 is not the content of Figure 1, please amend.

24. The content of lines 302 and 303 is not the content of Figure 2, please amend.

25. In 3.1“the malting quality of barley genotypes. ”in the last sentence should be suggested  to be amended to “the malting quality of  barley with different genotypes.”

26. Line 200~201,“There was significant effect of growing season on grain protein content. ”This sentence should be followed by significance at level.

27. Line 247: before the word “activity” should state the relevant substance.

28. Line 270~271: “Malt beta-amylase activity was significantly influenced by both the genotype and growing year but not by genotype x year interaction.” How is it affected by the growing year, please explain with specific data.

29. Please check whether the correlation data 76.3 % is correct with the above chart.

30. Line 173 “Correlation between different grain and malt quality traits.” It will look better if the picture is complete.

31. Line 299, in the “Figure 3. Principal component analysis (PCA) of barley genotypes based on the 13 traits (variables).” The font in the picture is changed to  The Times New Roman.

32. Line 168: note the serial number ordering, Pearson correlation (Figure 2) and PCA (Figure 3).

33. Line 230: no space between 6.1 and %, note check full text.

Discussion

34. The format of "et al" in line 331 is inconsistent with the preceding text, please check and amend.

35. The last two paragraphs of the Discussion section are poorly referenced, please add.

36. The format of "et al" in line 416 is inconsistent with the preceding text, please check and amend.

37. Line 834-835, “DWRUB 64 is a six rowed variety and has hot water extract value of 80.1%, which is exceptional. Higher HWE value in DWRB 91 despite exceptionally higher thousand grain weight and higher grain beta glucan content points to role of several other factors governing the final malt extract values.” Lack of literature.

38. Line 368~369,“Ram and Verma [22] have reported large variation in grain beta glucan content in the barley genotypes ranging from 2.8 to 7.1%.” Write a single author when quoting from the paper.

39. Line 390~392,“Verma and Sarkar [23] studied the malting quality of 72 Indian barley varieties and found positive correlation between test weight and hot water extract. ”Write a single author when quoting from the paper.

Conclusions

40. In line 430 there is an extra space before "The", please delete.

References

41. There is an error in the first article of documentation format, please correct it.

42. The format of the person's name in the second reference is incorrect, please correct.

43. Extra years in references 4, 7 and 8, please amend.

44. Some journal names in the references are not in italics, please change them.

45. Some references are in wrong format. For example: line 489, line 507, line 509

46. Line 463: journal name does not need to be bolded.

Reviewer 2 Report

The manuscript "Utilization of grain physical and biochemical traits to predict malting quality of barley (Hordeum vulgare L.) under sub-tropical climate" is quite interesting. The authors touch upon the important topic of studying barley for the brewing qualities of grain.

At the same time, the manuscript has some shortcomings.

In the abstract, the authors talk several times about guality traits, but don’t name them. This needs to be corrected.

In the research methods, the authors give a list of lines, without explaining why they took them, on what grounds they selected them, and what are these lines like? Why was there several six-row lines among the two-row lines of barley? The area of the plots and their design, which was used in this experiment, are not indicated?

Numbering of tables in the text is wrong. In table 1, the average yield for line No. 14 is not correct.

Line 175 Effect of growing year on guality traits. It is better and logical to rearrange - first discuss the effect of the quality traits on each other, and then the effect of growing year on them.
